# Roles of Small RNAs in Virus-Plant Interactions

**DOI:** 10.3390/v11090827

**Published:** 2019-09-05

**Authors:** Baogang Zhang, Wenji Li, Jialin Zhang, Lu Wang, Jianguo Wu

**Affiliations:** Vector-borne Virus Research Center, State Key Laboratory of Ecological Pest Control for Fujian and Taiwan Crops, Fujian Province Key Laboratory of Plant Virology, Institute of Plant Virology, Fujian Agriculture and Forestry University, Fuzhou 350002, China

**Keywords:** small RNAs, microRNAs, short interfering RNAs, symptom induction, resistance

## Abstract

Small RNAs (sRNAs), including microRNAs (miRNAs) and short interfering RNAs (siRNAs), are non-coding but powerful RNA molecules of 20–30 nucleotides in length. sRNAs play crucial regulatory roles in diverse plant biological processes. Recently, many studies on sRNAs have been reported. We summarize new findings of sRNAs in virus-plant interactions to accelerate the function analysis of sRNAs. The main content of this review article includes three parts: virus-responsive sRNAs, function analysis of sRNAs in virus pathogenicity or host resistance, and some sRNAs-mediated underlying mechanisms in virus-plant interactions. New findings of sRNAs deepen our understanding about sRNAs’ roles, which might contribute to the design of novel control measures against plant viruses.

## 1. Introduction

Small RNAs (sRNAs), which are a group of non-coding small RNA molecules of approximately 20–30 nucleotides (nt) in length, are mainly divided into two subgroups: microRNAs (miRNAs) and small interfering RNAs (siRNAs). The first miRNA, lin-4, which regulated *lin-14* transcript via RNA-RNA interaction, was identified in 1993 [1]. After that, the discovery and functional studies on sRNAs have become one hotspot in the past 30 years. Several new species and families of sRNAs have been discovered. Some sRNAs are responsive to abiotic and biotic factors, and they exist in all tissues during the whole life, playing crucial regulatory roles in plant growth and development [2,3,4]. sRNAs play inactive regulatory roles in most plant species during the entire life cycle by sequence-specific regulation of genes, transposons, parasites. Virus diseases caused serious crop losses. In Asia, rice viruses, such as rice stripe virus (RSV), rice dwarf virus (RDV), rice grassy stunt virus (RGSV), rice black streaked dwarf virus (RBSDV), and southern rice black streaked dwarf virus (SRBSDV) have huge economic impacts [5]. Virus infection differentially regulates the accumulation levels of sRNAs, which alters virus pathogenicity or host resistance [6]. Here, we summarize some new findings on roles of sRNAs in virus-plant interactions.

## 2. The Biogenesis and Action Modes of sRNAs

Many sRNAs are identified by cloning and genetic methods. High throughput sequencing technology greatly accelerated this process. The latest released miRBase (v22) data registered 48,860 mature miRNAs in 271 organisms (http://www.mirbase.org, release 22) [7]. miRNAs are from miRNA genes (*MIRs*), the vast majority of which are located in intergenic regions [2]. *MIRs* are transcripted by DNA directed RNA polymerase producing pri-miRNAs with imperfect complementary fold-back secondary structures [2]. Subsequently, pri-miRNAs are processed by a processing complex to produce mature miRNAs. The processing complex contains a set of core proteins, including Dicer-like RNase III (DCL), Double strand RNA binding protein (DRB)/Hyponastic leaves (HYL), Serrate (SE), and RNA methytransferase Hua enhancer (HEN) [8]. Among them, DRB/HYL protein is responsible for binding double strand RNAs, and DCL cuts double strand RNAs, and HEN is responsible for the methylation of duplex molecules to stabilize miRNAs [9]. In *Arabidopsis thaliana*, there are ten DCLs, four DRBs, one SE, and one HEN. Components in the same family played redundant or specific functions. For example, mature miRNAs of 21-, 22-, or 24-nt in length are responsible for DCL1/4, DCL2, and DCL3, respectively. siRNAs are divided into endogenous siRNAs and exogenous siRNAs. Endogenous siRNAs consist of five subcategories: trans-acting siRNAs (ta-siRNAs), natural sense-antisense transcript siRNAs (nat-siRNAs), repeat-associated siRNAs (ra-siRNAs), long siRNAs (lsiRNAs), and heterochromatic siRNAs (hc-siRNA). The biogenesis of ta-siRNAs is directed by miRNAs. In *Arabidopsis thaliana*, miR173, miR390, and miR828 direct the cleavage of *TAS1*/*TAS2*, *TAS3*, and *TAS4*, respectively. The precursor of nat-siRNA is from overlapping regions of natural sense-antisense transcript pairs [10]. They can be induced by environmental clues. DCL1/3, SGS3 (suppressor of genesilencing), and RDR2/6 (RNA-dependent RNA polymerase) are required for double-strand RNAs processing [10,11]. lsiRNAs are 30-40 nt in length, whose biogenesis requires DCL1/4, HYL1, HEN1, RDR6, and Polymerase IV [12]. hc-siRNAs are derived from transposons or repeated sequences, and they can be transcripted by Polymerase IV; hc-siRNAs play roles in DNA and H3K9 methylation at homologous chromatin to maintain genome integrity [13]. The biogenesis of hc-siRNAs also requires processing complex, including RDR2, DCL3, and HEN1 proteins [2]. In addition to these endogenous siRNAs, plants may contain exogenous siRNAs derived from virus, bacteria, fungi, oomycetes, or other parasites in parasite-plant interactions. The biogenesis of virus-derived siRNAs (vsiRNAs) is analogous to endogenous sRNAs and also requires DCL and RDR proteins [14]. Six RDR proteins have been predicted in *Arabidopsis thaliana* [15,16], but only one or two RDR proteins have been predicted in rice and maize (*Zea mays* L.) [17,18]. Phylogenetic analysis suggests that three RDR gene copies are present in the eukaryotic ancestor, with RDRα clade members RDR1, RDR2, and RDR3 involved in antiviral immunity [19]. Recently, aminophospholipid transporting ATPase (*ALA1* and *ALA2*) have been demonstrated to be necessary for the biogenesis of vsiRNAs or endogenous siRNAs, which is the down-stream of RDRs in *Arabidopsis thaliana* [20].

The produced mature sRNA is selectively sorted into RISC complex with Argonaute (AGO) protein for the following modification of targets by mRNA degradation, translational inhibition, DNA and/or histone methylation [2,21,22]. sRNAs often regulate gene expression through DNA/histone methylation, mRNA translational inhibition or mRNA cleavage at either transcriptional gene silencing (TGS) or the posttranscriptional gene silencing (PTGS) level. For example, AGO proteins mainly mediated RNA interfering by mRNA cleavage. Ten AGOs are encoded in *Arabidopsis thaliana*. AGO1, AGO2, AGO5, AGO7, and AGO10 proteins are mostly assembled with 21-22-nt sRNAs, whereas the AGO4, AGO6, and AGO9 proteins are mainly assembled with 24-nt sRNAs. 21-nt miRNAs are incorporated into the AGO1 protein to cleave target mRNAs, while long miRNAs direct DNA methylation of their own loci by AGO4 in *Arabidopsis thaliana* [23]. According to 5′ end rules, AGO1, AGO2, AGO5, and AGO7 preferentially interact with 5′ uridine, 5′ adenosine, 5′ cytosine, and 5′ uridine miRNAs, respectively. However, on occasion, AGO1 and AGO2 act redundantly in miR408-mediated plantacyanin regulation [24]. In addition, AGO1 has diverse functions, such as chromatin remodeling, co-transcriptional splicing regulation, as well as RNA maturation [25]. The majority of 21-nt siRNAs direct mRNA cleavage, while 24-nt siRNAs mediate DNA or histone methylation. For example, target mRNA cleavage or DNA methylation mediated by 21-nt ta-siRNA requires DCL4, AGO1 for cleavage, or DCL1, AGO4/6 for DNA methylation [26]. In *Arabidopsis thaliana*, DCL1 is responsive for miRNAs’ biogenesis, DCL3 is responsive for endogenous siRNAs’ biogenesis, and DCL2 is responsive for viral siRNAs’ biogenesis [3]. In Figure 1 we can take a glance on the biogenesis and action modes of sRNAs.

## 3. Virus-Responsive sRNAs in Virus-Plant Interactions

Virus infection alters the accumulation levels of sRNAs in host plants. Firstly, the ratio of different size of sRNAs changed after virus infection. When virus infected hosts, 21- and 22-nt sRNAs of plants increased, while 24-nt sRNAs decreased. For example, sRNAs changed in the potato virus Y (PVY)-tobacco (*Nicotiana tabacum*) and *Beta macrocarpa*, mungbean yellow mosaic india virus (MYMIV)-blackgram (*Vigna mungo*) interactions [27,28]. In the interactions between cucumber green mottle mosaic virus-*Lagenaria siceraria*, RBSDV-maize (*Zea mays*), the size distribution of vsiRNAs showed similar phenomena [29,30]. However, size distribution of vsiRNAs of distinct viruses is not the same. 22-nt siRNAs of cotton leafroll dwarf polerovirus (CLRDV) are predominant in virus infected cotton (*Gossypium hirsutum*) leaves, and 21–24 nt long viral sRNAs have higher frequencies matching in the 3′ region of the viral genome [31]. Accordingly, the metabolism of sRNAs is changed in response to virus invasion. 

The profiles of sRNAs responsive to virus invasion are explored to gain insights into metabolism of sRNAs in virus infection. The expression of miR162 increases in response to infection of CLRDV in cotton [31]. miR444, a monocot-specific miRNA that is ubiquitously expressed in rice, wheat, barley, sorghum, and sugarcane, is induced by RSV, while miR528 and miR396 are down-regulated by RSV [32,33]. RSV invasion also alters the accumulation levels of other miRNAs, for example, RSV increases the accumulation levels of miR159a/b/e/f, 168, 394, 395, 398, 399, 160*, 171*, and 1425*, and decreases the accumulation levels of miR156, 166, 167, and 171 [34,35,36]. miR168 is induced by CymRSV, tobacco mosaic virus (TMV), potato virus X (PVX), tobacco etch virus (TEV), and other viruses [37]. In addition to miRNAs, siRNAs are affected by viral infection. siRNAs, called P5_3′ or P10_3′, are induced by RSV [34]. Different from vsiRNAs in virus-host interactions, a distinct class of viral activated siRNA (vasiRNA) is activated by RNA viruses. Most vasiRNAs are mapped to the exon region of host genes and rRNA, requiring DCL4/RDR1/AGO2 proteins for their biogenesis and action in *Arabidopsis thaliana* [38]. Infected by viruses in plants, the accumulation levels of miRNAs, endogenous siRNAs, and vsiRNAs changed in host plants. Accordingly, how does virus affect sRNAs’ accumulation in hosts? Some researches revealed that some viral proteins have been found to alter sRNAs’ accumulation when expressed alone in host plants. In virus-rice interactions, RDV infection induced OsRDRs, whereas RSV elevated expression of some specific DCLs and AGOs. Correspondingly, RDV-regulated miRNA*s and miRNAs are less than RSV [34]. Viral protein was identified to act as scaffold between host plant and pri-miRNAs. RSV-encoded NS3 was firstly identified as a RNA silencing suppressor reducing levels of siRNAs in tobacco, and it binds to siRNAs, short duplexes, and long ssRNAs [39]. However, in RSV-rice interactions, RSV-encoded nonstructural protein NS3 interacts with DRB1 protein to aid pri-miRNA processing in vivo accelerating miRNAs’ biogenesis [36]. Table 1 summarizes sRNAs that altered by virus infection. 

## 4. sRNAs-Mediated Viral Disease Symptoms

Viral infection normally causes developmental defects in vegetable or reproductive stages. These virus-caused abnormities can also be called disease symptoms, for example, virus infection changes stem height or number [30,35,42,47], leaf shape or color [29,35,36,42,47,48,49,50], size of root crowns [47], plant fertility rates [35], plant fruit [35], and even seriously causing the plant death [51]. Cucumber mosaic virus (CMV) 2b protein interacts with host catalase, CAT3, causing necrosis spots in *Arabidopsis thaliana* [49]. OsIAA10 causes dwarf phenotype and increases tillers by negatively regulating auxin signaling pathway, and it is degraded by TIR1-mediated protein degradation pathway. RDV-encoded capsid protein P2 causes viral symptoms by targeting OsIAA10 [47]. What the underlying forming mechanisms of viral symptoms is one longstanding and interesting scientific question. As reviewed by Pallas in 2011, hypersensitive response, R resistance genes, RNA silencing, plant hormones, and other unknown mechanisms contributed to pathological phenotypes [52]. Secondary siRNAs’ biogenesis is blocked by a suppressor, such as viral βC1 encoded by TYLCCNV satellite, which requires *Nbrgs-CaM* expression to decrease RDR6-mediated sRNAs’ production, and alters viral symptoms [53]. This indirectly revealed the crucial roles of sRNAs in the forming of viral symptoms. Increasing evidences indicated that sRNAs played pivotally regulatory roles in the induction of virus-induced disease symptoms. 

The accumulation levels of sRNAs alter plant leaf morphology. It was observed that a correlation between symptom severity and the accumulation levels of miR156, 160, 166, 169, and 171 [54]. miR156, miR160, miR164, miR319, miR390, and miR396 are involved in leaf morphogenesis, and miR160 interacted with miR165/166 to control leaf development [55]. It is known that miRNAs and ta-siRNAs play roles in regulating leaf development by a comprehensive network [56]. Virus can alter miRNAs’ accumulation, contributing to construct disease symptoms. Double mutant of miR159 (*mir159ab*) presents severe stunting and up-curled rosette leaves in *Arabidopsis thaliana* [42]. When infected by a mild subgroup II strain CMV-LS, *Arabidopsis thaliana* silencing of miR165 and miR166 shows viral symptoms that resemble those infected by the severe strain [42]. Hence, miR165/166 are involved in the viral symptoms. miR159 was proved to negatively regulate this viral symptom by targeting *MYB33* and *MYB65* [42]. The inhibition of osa-miR171b causes stunting and leaf yellowing symptoms, and there are less chlorophyll in rice [35]. Rice that was infected by RSV for 30 days displayed less osa-miR171b accumulation level, presenting the viral symptoms [35]. Further research reveals that osa-miR171b-mediated viral symptoms are dependent on its targets, OsSCL6-IIabc [35]. In addition to miRNAs, siRNAs also alter viral disease symptoms. The first identified case is a siRNA derived from the CMV Y-satellite (Y-sat) RNA. It targets tobacco chlorophyll biosynthetic gene (*Chll*), causing *Chll* mRNA silencing, which is solely responsive for the yellowing symptom in tobacco [57]. Additionally, Y-sat mediated target mRNA cleavage is specifically dependent on the 22-nt long siRNA sequence by using the RNA silencing machinery [44]. sRNAs alter the seeds filling rate and vigor. The suppression of miR1432 increases the grain filling rate and grain weight [58]. miR1432 targets Acyl-CoA thioesterase (*OsACOT*) regulating the metabolism pathway of fatty acid, auxin, and abscisic acid [58]. Under the same aging condition, the accumulation of osa-miR164c and osa-miR168 are related with the seed germination rates. The accumulation of miR168 is positively related with seed germination rates, while the accumulation of miR164c is negatively related with seed germination rates [4]. 

## 5. sRNAs-Mediated Pathogenic or Resistant Mechanisms

Plants displayed resistant or susceptible phenotypes based on the strength and effect of antiviral defenses against viruses. The resistant or susceptible phenotype is compared with the phenotype of control. We judge the pathogenic or resistant phenotypes according to the viral disease severity, which is assessed by viral symptoms and/or viral accumulation. Plant immunity to some pathogens is often referred to high resistance to some pathogens, presenting no obvious disease symptoms after being infected by these pathogens. Plant resistance to some pathogens is referred to slight damage or losses after infected by pathogens. Growth and development losses are not serious in crop plants with pathogen resistance. However, the susceptible phenotype of plant is referred to be sensitive to pathogens, which presents serious damage in plant growth and development, even host death. In virus-plant interactions, the severity of disease symptoms and viral particles’ accumulation are often used to assess the host compatibility to viruses, although they are not always completely relevant. Increasing evidence reveals that sRNAs play roles in pathogenic or resistant mechanisms. Firstly, according to bioinformatics prediction, miRNAs have strong potential for antiviral activity through the formation of miRNA–viral genome pairings [59]. The tumor protein (TCTP) is required for potato virus Y (PVY) successful infection in tobacco; however, vsiRNAs target *TCTP* gene triggering plant viral resistance in PVY-tobacco interactions [27]. miRNAs involved in immunity to various pathogens, including bacteria, fungus, virus and parasites [60,61,62,63,64,65,66,67,68,69]. Here, we focus on sRNAs-mediated viral pathogenic or resistant mechanisms. In 2014, Du et al. provided direct evidence revealing that the accumulation level of miR159ab was correctly associated with severity of viral disease symptoms after various virulence group of the same virus infection [42]. Viruses can enhance their toxicity via sRNAs. As for siRNAs, vsiR3114 of cotton leaf curl Multan virus (CLCuMuV) target cotton contig28334 to increase plant viral susceptibility [46].

Nicaise et al. (2014) have proposed four antiviral mechanisms: dominant resistance, recessive resistance, and RNAi- or hormone-mediated resistance [70]. As the RNA silencing pathway interacts with other antiviral mechanisms, it is hard to clearly distinguish RNA silencing-dependent or-independent antiviral mechanisms. The RNA silencing is still the most effective approach to suppress viral proliferation. For example, RISC targets viral nucleocapsid or movement proteins to block viral amplication [71]. RNA silencing is triggered by either endogenous or exogenous dsRNAs, and it requires DCLs, AGOs, DRBs, and RDRs [72]. RNA silencing that is directed by vsiRNAs requires some of DCLs, AGOs, DRBs, RDRs, and other components. At least four distinct RNA silencing pathways were discovered since 2008, and they were separately by DCL1-4 [72]. Knock-out or knock-down of RDRs enhances viral toxicity on *Arabidopsis thaliana*, tobacco, tomato, pepper, rice, and maize; conversely, the overexpression of RDRs protects tomato, tobacco, and pepper against viruses [15,16,17,18,73,74,75]. AGO1 is the key factor of RISC, and the hypomorphic*ago1* mutant exhibits increased susceptibility to various viruses [72,76]. In *Arabidopsis thaliana*, four DCLs, two AGOs, one DRB, and one RDR were identified. In virus-plant interactions, the plants take advantage of distant vsiRNAs to down-regulate viral RNAs/DNAs by RNA silencing [14]. As a counter defense, viruses use various suppressors to overcome RNA-silencing-mediated antiviral defense [14]. Viral suppressors could perturb sRNAs’ biogenesis to indirectly alter viral disease symptoms. Suppressors are classified into viral suppressors of RNA silencing (VSRs) and endogenous suppressors of RNA silencing (ESRs), according to their sources. Previously identified VSRs include Pns10, P6, NS3, 2b, and so on [36,39,77,78,79], and ESRs include Nbrgs-CaM [53,80]. Although the structural flexibility of miRNAs increases the versatility and regulatory potential, P19 has usurped this process by changing affinity between AGO10 and 22-nt miR168 [81]. RDV-encoded Pns10 down-regulates the expression level of RDR6 in *Arabidopsis thaliana* and tobacco [77]. The P6 protein is the suppressor of rice yellow stunt rhabdovirus (RYSV), blocking RDR6-mediated secondary siRNAs’ biosynthesis [78]. A DNA satellite that is associated with the geminivirus tomato yellow leaf curl China virus (TYLCCNV) encodes the βC1 protein, which induces the calmodulin-like protein Nbrgs-CaM in tobacco and *Arabidopsis thaliana* to suppress RDR6 expression, so Nbrgs-CaM was identified as an endogenous suppressor of RNA silencing [53,80]. Vice versa, these silencing components are regulated by sRNAs. AGO1 is the target of miR168, negatively regulated by miR168-directed mRNA cleavage. Though CymRSV infection induces the expression of miR168 and mRNA of AGO1 at the same time, the AGO1 protein significantly decreases in *Arabidopsis thaliana* [37]. In *Arabidopsis thaliana*, DCL1 mRNA is negatively regulated by miR162, which indicates the existence of a feedback loop in the biogenesis of miRNAs by miRNAs-mediated RNA silencing [43]. Figure 2 shows some recently identified cases about sRNAs that are involved in virus-plant interactions. To protect AGO1-mediated antiviral resistance, plants have evolved a decoy protein AGO18. AGO18 is induced by RSV and competes with AGO1 for miR168 to alleviate the degradation of AGO1a; it is dependent on the PAZ domain, but not the PIWI domain, which indicates that the protection relies on sRNAs’ binding function, but not silencing function [41]. In addition, AGO18 functions as a core regulator in different viral resistance pathways. In *Arabidopsis thaliana*, 22-nt isomiR168 could bind to AGO10, the homolog protein of AGO1, to keep AGO1’s homeostasis [81]. Accordingly, AGO10 can function as a miRNA locker to suppress the degradation of class III *HOMEDOMAIN-LEUCINE ZIPPER* (*HD-ZIP III*), which is crucial for SAM development, by sequestering miR166/165 in *Arabidopsis thaliana* [82,83]. Osa-miR444 enhances antiviral resistance by regulating RDR1-mediated RNA silencing pathway in rice. RDR1 positively regulates plant antiviral immunity, but its expression is repressed by MADS box proteins that directly bind to *RDR1* promoter regions. OsMADS23/27a/57 are targets of osa-miR444, thus osa-miR444 indirectly regulates RDR1’s homeostasis in plants [32]. In cotton, viral resistance against cotton leaf curl Burewala virus (CLCuBV) is also enhanced by miRNAs-mediated RNA silencing. Among them, Ghr-miR168 was predicted to be perfectly complementary to the C1, C3, C4, V1, and V2 of CLCuBV genome [45]. Hence, the miRNAs-mediated antiviral resistance is well studied; however, other sRNAs are less studied. 

sRNAs regulate *R* gene-mediated viral resistance. During the process of TMV infection of tobacco, 22-nt miR6019 directs the biogenesis of 21-nt secondary miR6020 with the help of RDR6 and DCL4; these two miRNAs function as a guide strand to cleave Toll and Interlecukin-1 receptor-NB-LRR receptor *N* genes, thereby negatively regulating TMV resistance [84]. Antiviral RNAi-defective2 (AVI2), a multispan transmembrane protein, was identified to be necessary for DCL2-mediated vsiRNAs’ biogenesis, and it plays a partially redundant role with AVI2H in RDR1-medaited the biogenesis of vsiRNAs and vasiRNAs [85]. CMV induces it and it is required for antiviral RNAi, the underlying mechanism needs further research [85]. The molecular mechanism is conserved in three *Solanaceae* species-tobacco, tomato, and potato, such as nta-miR482/*N*, nta-miR6019/*N*, nta-miR6020/*N*, sly-miR6022/*Cf9* (6-10), sly-miR6023/*Cf9* (6-10), sly-miR6024/*Tm2^2^*, sly-miR6026/*Tm2^2^*, stu–miR482b/*NL25*, stu–miR482b/*Ry*, stu–miR482c/*RB4* (4), stu–miR482d/*R2*(10+), stu-miR6024/*RX2*, stu-miR6024/*Rpn1-vnt1*, and stu-miR6026/*Rpn1-vnt1*.

There are other sRNAs-mediated antiviral mechanisms. Reactive oxygen species (ROS), generated during virus invasion, especially hydrogen peroxide, hydroxyl radicals, and superoxide peroxide, positively regulate antiviral immunity in plants [33,86]. Additionally, ROS burst is one the plant antiviral defense, enhancing rice viral resistance. Research in rice reveals that AGO18 competitively binds to miR528 to protect its target gene, L-*ascorbate oxidase* (*AO*) mRNA, boosting ROS-mediated resistance [33]. Although miR528 negatively regulates the ROS levels by targeting its target genes, such as *AO* and *SPL9*, it is suppressed by RSV. SPL9 is a squamosa promoter binding protein-like family member, which directly binds to miR528 promoter to active expression [33,87]. Antiviral immunity mediated by plasmodesmata and hydrogen peroxide may be correlated with the spread of RNA silencing signals [88]. Successful viral infection depends on various susceptibility factors, while within-host viral replication requires numerous host proteins; consequently, a functional deficiency in these factories may block or prevent viral invasion [89]. The function defect of P58^IPK^ leads to enhanced viral pathogenicity, therefore it functions as a susceptibility factor [51]. Several such factors were identified, including *eukaryotic translation elongation factor 1A* (*eEF1A*), the *SUMO-conjugating enzyme* (*SCE1)*, *heat shock protein 70* (*Hsp70*), and *essential for poteXvirus accumulation 1* (*EXA1*) [90,91,92,93]. Bendahmane et al. (2002) reported that a viral coat protein from potato virus X was recognized by the Rx protein in potato and triggered hypersensitive response in leaves [94]. Many avr-R protein pairs have been identified. The viral CP proteins from PVX, TCV, and CMV are recognized by Rx, HRT (HR to turnip crinkle virus), and RCY1 (Resistance to CMV (Y)) [95,96,97]. In addition, other viral components could be recognized by R proteins, such as cylindrical inclusion protein (CI)-TuRB01, cytoplasmic inclusion cistron-Rsv3, P3-TuMV, and so on [98,99,100]. Viral invasion differentially regulates phytohormones homeostasis, for example, RBSDV invasion increases the concentration of abscisic acid (ABA), cytokinins (CTKs), whereas decreases the concentration of indole-3-acetic acid (IAA), gibberellins (Gas), jasmonic acid (JA), and salicylic acid (SA) in rice [47,101,102]. Accordingly, hosts and viruses competitively regulate plant hormones. Growing evidence revealed that sRNAs-mediated hormones homeostasis played crucial functions in virus-plant interactions. Osa-miR319 is induced by rice ragged stunt virus (RRSV), which reduces *OsTCP21* expression and JA concentrations, and rice with low JA levels is more susceptible to viruses [40]. Recently, osa-miR393 overexpressed in rice presents more susceptibility to RBSDV by suppressing auxin recognization, in which JA content is decreased when infected by virus [103]. sRNAs-mediated methylation of viral DNA or RNA genomes or other fragments is a common defense strategy against DNA viruses [6]. Methylation is one of the viral defense responses. In CMV-tobacco interactions, the DNA methylation plays important roles in modulating viral clearance and resistance signaling [104]. RNA silencing suppressor 2b encoded by CMV did not change sRNAs production or transcription, but it reduced the plant methylation levels, and then increased viral pathogenicity [105]. In addition, genome editing that based on the CRISPR/Cas system can be used to protect hosts from viruses; this approach has been demonstrated to be effective antiviral strategy against DNA and RNA viruses in dicot and monocot plants [106,107].

## 6. Conclusions and Perspectives

Some viral diseases cause serious production and quality losses [5,30,108]. Viruses, which obligate intracellular pathogens, require fatty acids, carbon, energy, and other materials and cellular structures for replication in host cells, leading to the second largest crop losses [47,50,109,110]. However, viral diseases are rarely curable due to the lack of effective control measures. Different viruses may have variant pathogenic mechanisms, which require various management strategies [101]. Antiviral materials screened by genetic method for artificial breeding is an effective viral management strategy, and revealing antiviral mechanisms will accelerate the genetic breeding. With the development of sequencing technology, more and more sRNAs were identified [111]. Their crucial regulatory roles are gradually recognized. This review highlighted the functional roles and the underling mechanisms of miRNAs and siRNAs in virus-plant interactions, which may aid in the development of virus-resistant varieties.

sRNAs-mediated RNA silencing and gene expression modification can be used to manipulate plant immunity. Researches on sRNAs provide inspiration for crop breeding that is based on their own attributes and advantages. Firstly, plant miRNAs’ regulatory specificity provides a basis for artificial miRNA usage in agriculture [112]. sRNAs perform their specific functions at designated positions in accordance with complementary base pairing rules by cleavage, translation inhibition, and methylation, so they often act as precision-guide weapons. They play crucial regulatory functions in plant antiviral resistance by many different mechanisms, which enlarged the usage scale in technology. Secondly, sRNAs are short in length. It is a natural advantage in regulating plant development and immunity, because their small size made them flexible and diffusible. They can easily transfer from cell to cell, tissue to tissue, and even species to species. sRNAs shuttle between different species. sRNAs in plants are not only derived from the host itself, but also from its interacting species [113]. The fungal pathogen *Botrytis cinerea* was previously reported to transfer sRNAs into host plant cells to silence host immunity genes by hijacking the AGO1-mediated silencing mechanism [114]. A similar phenomenon was observed in the interactions between cotton and *Verticillium dahliae*. When *Verticillium dahliae* infects cotton plants, plants exported both miR166 and miR159 into the fungus to inhibit virulence gene expression [115]. Although vsiRNAs are detected in host plants, we think viruses could be targeted by plant sRNAs. Hence, we should further dig their potential functions by focusing on the different sRNAs interaction networks. Plant miR2911 encoded by honeysuckle directly targets virus in mammals, and plant miR168 can specifically target mammals low-density lipoprotein receptor adapter protein 1 [116,117]. Recently, the accumulation of specific virus gene is silenced by feeding with siR471 and siR519 transgenic lettuce of artificial interfering miRNAs in mice [118]. Accordingly, the shuttle of miRNAs in different species has given us inspiration that regulatory function of miRNAs are more powerful and they have potential application in the aspect of RNAi-mediated genetic therapy [118]. Thirdly, sRNAs-mediated regulation is fine-tuning of gene expression otherwise an on-off regulation. In plant breeding, a desirable trait is often balanced by some other unexpected characteristics. As superior fine-tuners of complex crop traits, miRNAs have intrinsic, unmatched advantages for the balancing of sharply conflicting features that are useful for crop improvement [119]. Plant fitness and stresses resistance are two main opposite traits in the crop breeding work, so how to take care of both sides is an important question for the breeders taking into consideration. The common way to solve the problem is to sacrifice some fitness penalty to enhance their resistance to various stresses. However, some researchers have solved the problem in a different way. IPA1 and OsSPL7, the target of miR156, enhances the rice resistance against bacterial blight, *Xanthomonas oryzae* pv. *oryzae* (Xoo), but decreases the grain yield in rice. However, induced the expression of IPA1 driven by the OsHEN1 promoter increases both Xoo resistance and grain yield [69]. This study provides a new approach for us to take stresses resistance and yield into consideration, which also can be used for antiviral breeding. Feedback regulation to alleviate an over-accumulated defense response is important for plant development because a fitness cost is associated with response to pathogen infection. It is possible that sRNAs are involved in this feedback regulation [84]. Based on these data, to reduce the fitness and production costs, the sRNAs-mediated underlying mechanisms need to be further investigated. Additionally, we think that sRNAs’ expression levels can be manipulated to the suitable, valuable agronomic traits in food and economic crops.

## Figures and Tables

**Figure 1 viruses-11-00827-f001:**
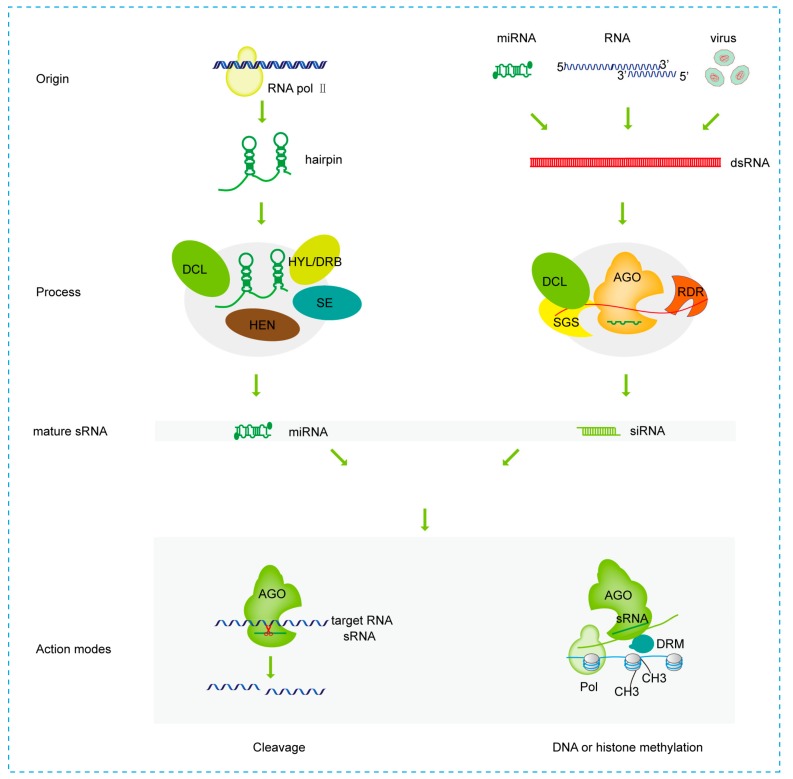
The schematic diagram of biogenesis and action modes of small RNAs (sRNAs). *MIR* are transcripted by RNA polymerase II producing pri-miRNAs with hairpin structure. Mature miRNAs are produced by processing complex including Dicer-like RNase III (DCL), Hyponastic leaves/Double strand RNA binding protein (HYL/DRB), Serrate (SE), and Hua enhancer (HEN) proteins. Mature miRNA is incorporated into action mode complex to direct the target silencing by cleavage or methylation. Based on the difference in origin, siRNAs are divided into endogenous or exogenous siRNAs. Viral genome or replication intermediates all can form hairpin structure. Additionally, host native or viral siRNA-directed double strand RNAs are the origin of sRNAs’ biogenesis. Double strand RNAs are processed by processing complex, which mainly include DCL, Argonaut (AGO), Suppressor of genesilencing (SGS), and RDR (RNA-dependent RNA polymerase) proteins to produce siRNAs. Mature siRNA is incorporated into action mode complex to direct the target silencing by cleavage or methylation at transcriptional or post transcriptional level.

**Figure 2 viruses-11-00827-f002:**
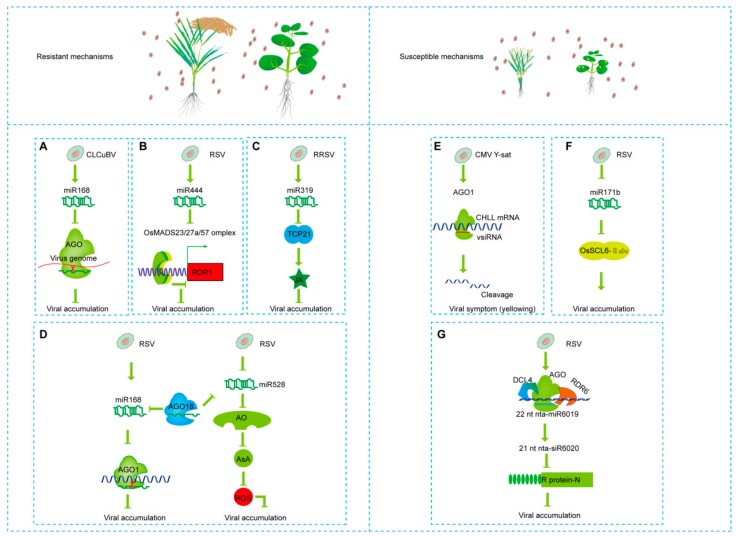
sRNAs-mediated viral pathogenic and resistant mechanisms. Left column of the figure is resistant mechanisms (**A**–**D**), right column of the figure is susceptible mechanisms (**E**–**G**). (**A**) Cotton leaf curl Burewala virus (CLCuBV) induces the expression of miR168. The miRNA can directly target virus genome, enhancing viral resistance. (**B**) Rice stripe virus (RSV) induces the accumulation of miR444. MADS23/27a/57 are targets of miR444, and they can bind to the promoter of RDR1 (RNA-dependent RNA polymerase), inhibiting the expression of RDR1. So, miR444 indirectly triggered RDR1-mediated viral resistance. (**C**) Plant hormone JA mediated viral resistance. Responsive to rice ragged stunt virus (RRSV) infection, miR319 is induced and negatively regulates TCP21, and further suppresses JA-mediated viral resistance. (**D**) In rice, AGO18 functions as a miRNA locker, competitively binding to miR168 or miR528 to increase AGO1- or ROS-mediated viral resistance. (**E**) CMV Y-Sat prompts the production of siRNAs by inducing the expression of AGO1. These siRNAs direct the cleavage of *Chll* (chlorophyll biosynthetic gene) mRNA. Tobacco *Chll* is responsible for the chlorophyll biogenesis, so, the plant displayed viral disease symptoms. (**F**) RSV infection suppresses the accumulation level of miR171b, which targets OsSCL6-IIabc to accelerate the accumulation of virus. (**G**) In tobacco, 21-nt nta-miR6020, which is derived from 22-nt miR3019, targets R protein, suppressing domain resistance.

**Table 1 viruses-11-00827-t001:** The list of sRNAs involved in virus-plant interactions.

sRNA Name	Putative Target(s)	Virus(es)	Putative Pathway(s)	Reference
Osa-miR164	OsNAC	Rice ragged stunt virus (RRSV)	leaf morphogenesis	[40]
Osa-miR168	OsAGO1a	Rice stripe virus (RSV)	RNA silencing	[41]
Osa-miR171b	OsSCL6-IIa/b/c	RSV	chlorophyll biosynthesis	[35]
Osa-miR319	Teosinte branched/cycloidea/pcf	RRSV	JA biosynthesis and signaling pathway	[40]
Osa-miR444	OsMADS23/27a/57	RSV	viral immunity	[32]
Osa-miR528	OsAO	RSV	L-ascorbic acid oxygen	[33]
Tae-miR164	TaNMO	Rice black streaked dwarf virus (RBSDV)	unknown	[40]
Tae-miR319	TaPCF8	RBSDV	JA biosynthesis and signaling pathway	[40]
Ath-miR159	AtMYB33/55	Cucumber mosaic virus (CMV)	unknown	[42]
Ath-miR162	AtDCL1	Turnip yellow mosaic virus TuMV	miRNAs’ biogenesis	[43]
Ath-miR165/166	unknown	CMV-LS	unknown	[42]
Nta-miR168	AGO1	CymRSV, crTMV, PVX, TEV	antiviral mechanism	[37]
Y-Sat	chlorophyll biosynthetic genes	CMV Y-Sat	chlorophyll biogenesis	[44]
Gh-miR162	GhDCL2	cotton leafroll dwarf polerovirus (CLRDV)	pathogenic mechanism	[31]
Gh-miR168	cotton leaf curl Burewala virus (CLCuBV) genome	CLCuBV	antiviral mechanism	[45]
Gh-miR395ad	CLCuBV genome	CLCuBV	antiviral mechanism	[45]
vsi3114	Contig28334	CLCuD	pathogenic mechanism	[46]

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
