# Peer review of "Roles of Small RNAs in Virus-Plant Interactions"

_viruses, 2019, doi:10.3390/v11090827_

Round 1

Reviewer 1 Report

The revised manuscript has been much improved. I have no concerns to publish this review.

Author Response

Thanks for your review.

Reviewer 2 Report

The manuscript reviews the roles of small RNAs in virus-plant interactions.  The manuscript does a good job of distinguishing groups of sRNAs, their biogenesis, modes of action, and their roles in suppressing plant defenses against viruses, and vice-versa.  The manuscript overall is well written, but there are places where the sub-headings are too long, which could be addressed by providing more sub-headings within each section.  The manuscript could also use more tables.  In addition, the manuscript has several grammatical errors that need to be addressed prior to being accepted for publication. The minor issues are listed below.

The title should be amended:  It should be ‘virus-plant’ interactions.

Line 25.  Should be ’30 years’.  Replace ‘Some’ with ‘Several’

Line 25. What to members mean here? Species? Please clarify.  

Lines 29-32. Why just focus on viruses from Asia and only from rice?

Line 35. Second subheading: This information could be presented in different sub-headings

classification Biogenesis Mode of action

Line 41.  Please replace ‘were’ with ‘are’.

Line 80.  Should be ‘adenosine’

Line 81.  Should be ‘act’ instead of ‘acted’

Line 97.  Should be sRNAs’ biogenesis.

Lines 104-105. Virus names here and elsewhere should not be italicized, unless the names mention their taxonomic status. Line 137, subheading 4.  A lot of the information presented here could use a table as well. 

Lines 180-185. This information is confusing.  If the authors are characterizing phenotypes based on degrees of resistance and sRNAs expression that should be explicitly stated.  

Line 214, It should be ‘as a counter defense, virus use’.

Line 233. Envolved? Please correct.

Subheading 5. This section is too long as well, and could be broken up.  

Line 303. Please replace ‘Susceptible’ with ‘Susceptibility’.

Line 314. Please delete the opening sentence.

Lines 343-44. Please replace ‘mammalian’ with ‘mammals’.

How would sRNAs be affected by environment? Can you comment on their expression stabilities with abiotic stresses such as heart and drought tolerance?

Author Response

Response to Reviewer 2 Comments

The manuscript reviews the roles of small RNAs in virus-plant interactions.  The manuscript does a good job of distinguishing groups of sRNAs, their biogenesis, modes of action, and their roles in suppressing plant defenses against viruses, and vice-versa.  The manuscript overall is well written, but there are places where the sub-headings are too long, which could be addressed by providing more sub-headings within each section.  The manuscript could also use more tables.  In addition, the manuscript has several grammatical errors that need to be addressed prior to being accepted for publication. The minor issues are listed below.

The title should be amended:  It should be ‘virus-plant’ interactions.

Response 1: Thanks for your advice. We replaced ‘viruses-plant’ with ‘virus-plant’ in title and the whole text.

Line 25.  Should be ’30 years’.  Replace ‘Some’ with ‘Several’

Response 2: Thanks for your advice. We replaced ‘30 year’ with ’30 years’ in line 25 of the former manuscript. And we replaced ‘some’ with ‘several’ in line 25.

Line 25. What to members mean here? Species? Please clarify.  

Response 3: Thanks for your question. We deleted ‘and members’ in the text.

Lines 29-32. Why just focus on viruses from Asia and only from rice?

Response 4: Thanks for your question. Firstly, rice, as a model plant of monocots, is studied extensively. Secondly, rice is one of the most important crops grown in Asian, especially in China. And viral diseases in rice caused serious economic losses in history. So we focus on rice viruses in Asian.

Line 35. Second subheading: This information could be presented in different sub-headings classification Biogenesis Mode of action

 Response 5: Thanks for your advice. In this manuscript, the part of biogenesis and action model of sRNAs is necessary for the latter part of the text, although it is not the focus of discussion to this review. So we put these two contents in the same subheading.

Line 41.  Please replace ‘were’ with ‘are’.

Response 6: Thanks for your advice. We replaced ‘were’ with ‘are’ in line 41.

Line 80.  Should be ‘adenosine’

Response 7: Thanks for your advice. We replaced ‘adenosin’ with ‘adenosine’ in line 80.

Line 81.  Should be ‘act’ instead of ‘acted’

Response 8: Thanks for your advice. We replaced ‘acted’ with ‘act’.

Line 97.  Should be sRNAs’ biogenesis.

Response 9: Thanks for your advice. We replaced ‘sRNAs biogensis’ with ‘sRNAs’ biogenesis’ in line 97.

Lines 104-105. Virus names here and elsewhere should not be italicized, unless the names mention their taxonomic status. Response 10: Thanks for your advice. We reselected normal fonts of these virus names, and we checked the whole text.

Line 137, subheading 4.  A lot of the information presented here could use a table as well. 

Response 11: Thanks for your advice. We summarized the contents of subheading 4 and 5 in Figure 2, so if we add a table here, the manuscript will present duplicate contents.

Lines 180-185. This information is confusing.  If the authors are characterizing phenotypes based on degrees of resistance and sRNAs expression that should be explicitly stated.  

Response 12: Thanks for your question. Researchers assessed the level of disease damage according to the disease index. In addition, we also assessed the disease level by quantifying the amount of pathogens by some molecular methods.

Line 214, It should be ‘as a counter defense, virus use’.

Response 13: Thanks for your advice. We replaced ‘viral used’ with ‘viruses use’ in line 214.

Line 233. Envolved? Please correct.

Response 14: Thanks for your advice. We replaced ‘envolved’ with ‘evolved’ in line 233.

Subheading 5. This section is too long as well, and could be broken up.  

Response 15: Thanks for your advice. In the war between plant and pathogens, they are co-evolving. The phenomenon was described by Zig-zag model. So, putting these two parts in the same subheading contributes to understanding the text for authors.

Line 303. Please replace ‘Susceptible’ with ‘Susceptibility’.

Response 15: Thanks for your advice. We replaced ‘susceptible’ with ‘susceptibility’ in line 303.

Line 314. Please delete the opening sentence.

Response 16: Thanks for your advice. We deleted the opening sentence in line 314.

Lines 343-44. Please replace ‘mammalian’ with ‘mammals’.

Response 17: Thanks for your advice. We replaced ‘mammalian’ with ‘mammals’ in line 343-44.

How would sRNAs be affected by environment? Can you comment on their expression stabilities with abiotic stresses such as heart and drought tolerance?

Response 18: Thanks for your question. sRNAs are regulated by environment clues, such as drought or hurt. In the review article called ‘MicroRNAs and their regulatory roles in plant-environment interactions (Song et al., 2019)’, authors reviewed that miRNAs were responsed to various environment clues. According to our unpublished data, we find that miRNAs are regulated by biotic or abiotic stimulations in a time period. And sRNAs will keep a dynamic equilibrium state.

This manuscript is a resubmission of an earlier submission. The following is a list of the peer review reports and author responses from that submission.

Round 1

Reviewer 1 Report

The main comment of this reviewer is that frequently the authors choose to dump a few references without digesting for the reader what the previously published work means. This make the manuscript at time more a collection of references than a review paper. As such, and although the authors themselves claim that such a review "should aid in the development of virus-resistant varieties", they came short of that goal.

This reviewer could not find the references 1, 22,49, 88 and 96 in the text. 

This reviewer calls attention for some of the citations in the text which do not seem to correspond to the work published.

Finally this reviewer encloses an annotated pdf of the manuscript to convey all the queries and suggestions to the authors.

Reviewer 2 Report

This review focuses small RNAs that function in plant–virus interactions. The scope of the manuscript is broad: authors tried to divide into several sections according to the functions, but the subtitle of each section is not necessarily match with the content. For example, “susceptible and resistant phenotype” means the phenotype of individual plants that do or do not allow virus multiplication, but the text has nothing to do with it; “immunity” is a word for the nature of host plants that restricts virus multiplication, but the text includes virus infection strategies. Despite many small RNAs and related proteins appears, explanations of each function are not organized or detailed. In addition, the manuscript contains many grammatical errors. So overall readability is poor as a review article for non-specialist readers.